# The proposed channel-enzyme transient receptor potential melastatin 2 does not possess ADP ribose hydrolase activity

**Iordan Iordanov[1,2], Csaba Mihályi[1,2], Balázs Tóth[1,2], László Csanády[1,2]***

[1]Department of Medical Biochemistry, Semmelweis University, Budapest, Hungary; [2]MTA-SE Ion Channel Research Group, Semmelweis University, Budapest, Hungary

**Abstract** Transient Receptor Potential Melastatin 2 (TRPM2) is a $Ca^{2+}$-permeable cation channel essential for immunocyte activation, insulin secretion, and postischemic cell death. TRPM2 is activated by ADP ribose (ADPR) binding to its C-terminal cytosolic NUDT9-homology (NUDT9H) domain, homologous to the soluble mitochondrial ADPR pyrophosphatase (ADPRase) NUDT9. Reported ADPR hydrolysis classified TRPM2 as a channel-enzyme, but insolubility of isolated NUDT9H hampered further investigations. Here we developed a soluble NUDT9H model using chimeric proteins built from complementary polypeptide fragments of NUDT9H and NUDT9. When expressed in *E.coli*, chimeras containing up to ~90% NUDT9H sequence remained soluble and were affinity-purified. In ADPRase assays the conserved *Nudix-box* sequence of NUDT9 proved essential for activity ($k_{cat}$~4-9s$^{-1}$), that of NUDT9H did not support catalysis. Replacing NUDT9H in full-length TRPM2 with soluble chimeras retained ADPR-dependent channel gating ($K_{1/2}$~1-5 μM), confirming functionality of chimeric domains. Thus, TRPM2 is not a 'chanzyme'. Chimeras provide convenient soluble NUDT9H models for structural/biochemical studies.

***For correspondence:** csanady. laszlo@med.semmelweis-univ.hu

**Competing interests:** The authors declare that no competing interests exist.

## Introduction

Transient receptor potential melastatin 2 (TRPM2) belongs to the TRP protein family and is predominantly expressed in brain neurons, spleen, bone marrow, leukocytes, phagocytes, pancreatic β cells, and cardiomyocytes (*Nagamine et al., 1998*; *Perraud et al., 2001*; *Togashi et al., 2006*). It forms $Ca^{2+}$-permeable nonselective cation channels which are activated under oxidative stress, and contribute to the $Ca^{2+}$ signals that trigger cell migration and chemokine production in immune cells (*Yamamoto et al., 2008*) and insulin secretion in the pancreas (*Uchida et al., 2011*). Under pathological conditions, such as post-ischemic reperfusion injury (*Hara et al., 2002*; *Kaneko et al., 2006*) and potentially in Alzheimer's disease (*Fonfria et al., 2005*), TRPM2 activity leads to $Ca^{2+}$ dysregulation and cell death. On the other hand, reduced activity of TRPM2 might be linked to amyotrophic lateral sclerosis and Parkinson's disease dementia (*Hermosura et al., 2008*), as well as bipolar disorder (*McQuillin et al., 2006*).

TRPM2 forms homotetramers with a transmembrane architecture similar to that of voltage-gated cation channels (*Figure 1A*). In addition, it contains a large cytosolic N-terminal domain of unknown function specific to the TRPM subfamily, and a C-terminal coiled-coil region (CCR, *Figure 1A*) followed by a small cytosolic domain (NUDT9H). The NUDT9H domain exhibits a high degree of sequence similarity (~35% identity, ~45% similarity) to the soluble, $Mg^{2+}$-dependent mitochondrial pyrophosphatase NUDT9 (*Shen et al., 2003*), which catalyzes the hydrolysis of adenosine 5'-diphosphoribose (ADPR) into AMP and ribose 5-phosphate (R5P) (*Figure 1B and C*). TRPM2 is activated by co-application of cytosolic ADPR and $Ca^{2+}$ (*Perraud et al., 2001*; *Sano et al., 2001*; *Csanády and Törocsik, 2009*), and the ADPR ligand was indeed shown to bind to the NUDT9H domain of the

**eLife digest** Ion channels are proteins that allow specific charged particles to move across the membranes of cells – for example to travel in or out of a cell, or between different parts of the same cell. Almost all ion channels are gated, meaning that they can open and close in response to different signals. For instance, so-called ligand gated channels open in response to binding of some small molecule, known as the "ligand". A small number of channel proteins are also enzymes, meaning that they are able to catalyze chemical reactions, and these channel-enzymes are often referred to as "chanzymes".

TRPM2 is an ion channel found in humans that opens when a small molecule called ADPR binds to a portion of the channel inside the cell. This channel is also thought to be a chanzyme because the part that binds to ADPR is similar to an enzyme called NUDT9. The NUDT9 enzyme converts ADPR into two other chemicals. When studied in biochemical assays, the enzyme-like part of TRPM2 – which contains a segment called a "Nudix box" – appeared to act in the same way, although this activity was not linked to the opening and closing of the TRPM2 channel.

Iordanov et al. set out to re-examine whether TRPM2 is actually an enzyme by comparing the activity of NUDT9 to the enzyme-like part of TRPM2. To test an enzyme's activity, it typically needs to be dissolved in water. However, the enzyme-like part of TRPM2 does not dissolve, and so it could not be tested directly. Instead, Iordanov et al. identified which parts of TRPM2 make it insoluble and replaced them with the equivalent parts from NUDT9 to create several new proteins. For all the proteins tested, only those with the Nudix box from NUDT9 were active enzymes, while those with the Nudix box from TRPM2 were not.

Iordanov et al. conclude that TRPM2 is a ligand gated channel and not a chanzyme, and that the experimental conditions used in previous biochemical assays, and not TRPM2 activity, led to the breakdown of ADPR. Finally, the TRPM2 channel is involved in cell damage following heart attacks or stroke and may contribute to Alzheimer's disease, Parkinson's disease and bipolar disorder as well. As such, knowing how TRMP2 behaves could guide efforts to develop new drugs for these illnesses.

channel (*Grubisha et al., 2006*). Based on its crystal structure, the mitochondrial enzyme is composed of two subdomains, termed Cap and Core (*Figure 1C*, *blue* and *brown*). The Cap subdomain facilitates binding of the substrate ADPR, whereas the Core subdomain binds the substrate and catalyses its cleavage to AMP and R5P (*Shen et al., 2003*; *Perraud et al., 2003*). The Core subdomain harbors the *Nudix* (nucleoside diphosphate linked moiety X) box - a conserved motif (*Bessman et al., 1996*) responsible for recognition and hydrolysis of ADPR, present in a variety of phosphohydrolases (*Gabelli et al., 2001*; *Kang et al., 2003*; *Gabelli et al., 2007*; *Huang et al., 2008*; *Messing et al., 2009*; *Boto et al., 2011*) including both NUDT9 and NUDT9H (*Perraud et al., 2001*).

The strong similarity between NUDT9 and NUDT9H raised the intriguing possibility that NUDT9H itself might act as an enzyme with ADPR hydrolase (ADPRase) activity. Indeed, previous work reported weak enzymatic activity for isolated NUDT9H (*Perraud et al., 2001*, *2003*), compatible with the slow timescale of channel gating (*Csanády and Törocsik, 2009*), classifying TRPM2 among channel-enzymes ('chanzymes'). This small group of proteins, in which a single polypeptide chain forms both a transmembrane ion pore and a catalytically active domain, includes two other members of the TRPM subfamily (TRPM6 and TRPM7), and the Cystic Fibrosis Transmembrane Conductance Regulator (CFTR) chloride ion channel. In CFTR catalytic (ATPase) activity is strictly coupled to channel gating, which therefore follows a nonequilibrium cyclic mechanism (*Csanády et al., 2010*). In TRPM7 catalytic (phosphotransferase) activity is not linked to pore gating (*Matsushita et al., 2005*), but instead serves to regulate expression of TRPM7-dependent genes (*Krapivinsky et al., 2014*). In the case of TRPM2 the physiological role of the proposed ADPRase activity has not yet been clarified, but we have previously shown that any hydrolytic activity of NUDT9H is not coupled to gating of the TRPM2 channel itself (*Tóth et al., 2014*). Furthermore, our observation of intrinsic instability of several pyridine dinucleotides at basic pH (*Tóth et al., 2015*) raised the possibility of a similar

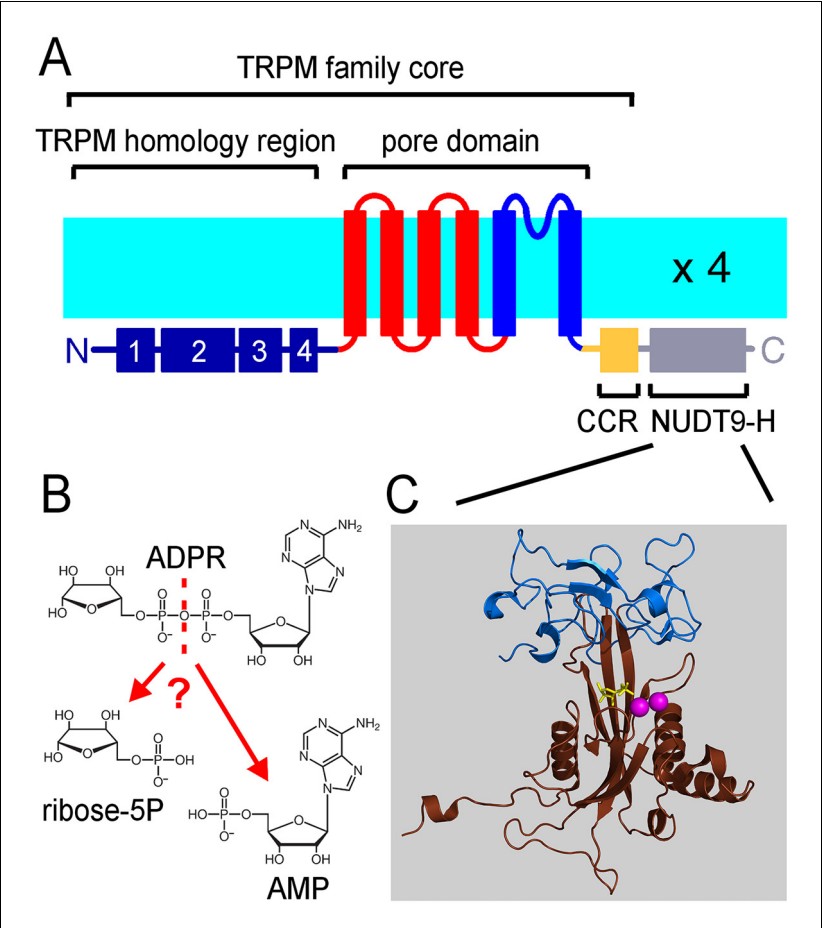

**Figure 1.** The ligand-binding NUDT9H domain in the context of TRPM2. (**A**) Topology map of a TRPM2 monomer, showing the N-terminal TRPM homology region (*dark blue*), the six transmembrane helices which form the pore domain (*red* and *blue*), and a C-terminal short coiled-coil region (CCR, *yellow*) followed by the NUDT9H domain (*gray*) which is homologous to the soluble mitochondrial ADPRase NUDT9. (**B**) Hydrolysis of ADPR to AMP and R5P: reaction catalyzed by mitochondrial NUDT9 and reportedly also by the TRPM2 NUDT9H domain. (**C**) Crystal structure of mitochondrial NUDT9 (PDB ID: 1QVJ) illustrating Cap (*blue*) and Core (*brown*) subdomains of the protein, with the hydrolysis product R5P (*yellow*) and two $Mg^{2+}$ ions (*magenta*) bound.

instability for ADPR, which might lead to false positive results when assessing the hydrolytic activity of a putative slow ADPRase enzyme. It thus remained a question whether the NUDT9H domain truly exhibits any enzymatic activity, not coupled to TRPM2 gating. One way to address this would be to further analyze the TRPM2 ADPR-binding domain in isolation. However, in our hands NUDT9H proved to be extremely insoluble (*Tóth et al., 2014*), despite its high degree of sequence similarity to highly soluble NUDT9. For this reason we first constructed a series of chimeric proteins built from complementary polypeptide fragments of the two homologous proteins, and correlated solubility with NUDT9H sequence content. This approach allowed us to pinpoint the NUDT9H segments responsible for its insolubility, and to purify and investigate soluble chimeric proteins which contain up to ~90% of the native NUDT9H sequence. We then investigated enzymatic activities of such soluble chimeras in isolation, as well as their capacities to substitute for native NUDT9H in conferring ADPR-dependent gating to the full-length TRPM2 channel.

## Results

### C-terminal residues of NUDT9H are responsible for its insolubility

In a previous attempt to purify and biochemically characterize the TRPM2 NUDT9H domain in isolation we found it to be essentially insoluble, unlike its mitochondrial homologue NUDT9 (*Tóth et al., 2014*). Because of this unexpected difference we further investigated the two proteins in an attempt to obtain a soluble analogue of NUDT9H. Prompted by the strong similarity between them (*Figure 2*), we generated chimeric constructs by swapping homologous segments between NUDT9 and NUDT9H. The crystal structure of mitochondrial NUDT9 ([*Shen et al., 2003*]; PDB ID: 1Q33) identified two subdomains - a 'Cap' and a 'Core' (*Figure 1C*, *blue* and *brown*). Assuming structural similarity between NUDT9 and NUDT9H, these two structurally distinct regions of NUDT9 provided the basis for generating the first chimeric constructs (Chi1/1A and Chi2/2A), obtained by exchanging the Cap and Core segments of the two native proteins (*Figure 3A*). The NUDT9-Cap/NUDT9H-Core chimeras Chi1 and Chi1A were insoluble, whereas the NUDT9H-Cap/NUDT9-Core chimeras Chi2 and Chi2A were soluble, and Chi2 could be purified by Ni-affinity chromatography (*Figure 3—figure supplement 1*). The low solubility of NUDT9H is thus caused at least in part by its Core subdomain.

The soluble Chi2 contains only ~41% of NUDT9H sequence (*Table 1*) and, notably, the catalytically important *Nudix* box in Chi2 originates from mitochondrial NUDT9. To obtain soluble constructs which more closely resemble NUDT9H, the point of transition from NUDT9H- to NUDT9-sequence was advanced further downstream. For the next construct, Chi3 (*Figure 3A*), this switch point was propagated about halfway through the Core segment, immediately after a well aligned, conserved stretch of amino acids (*Figure 2*) which forms a reentrant loop structure, pivotal for anchoring the Cap to the Core segment in NUDT9. A small fraction of this Chi3 chimera, which contains ~77% of NUDT9H sequence (*Table 1*), including its *Nudix* box, remained soluble and could be purified. We then advanced the transition point further downstream, about halfway through the remainder of the NUDT9 sequence in Chi3. This new construct (Chi4) is formed by ~91% of NUDT9H sequence (*Figure 2*, *Figure 3A and B*), but a small amount of it still remained in the soluble fraction which allowed us to purify it. The low yields of soluble Chi3 and Chi4 could be robustly increased

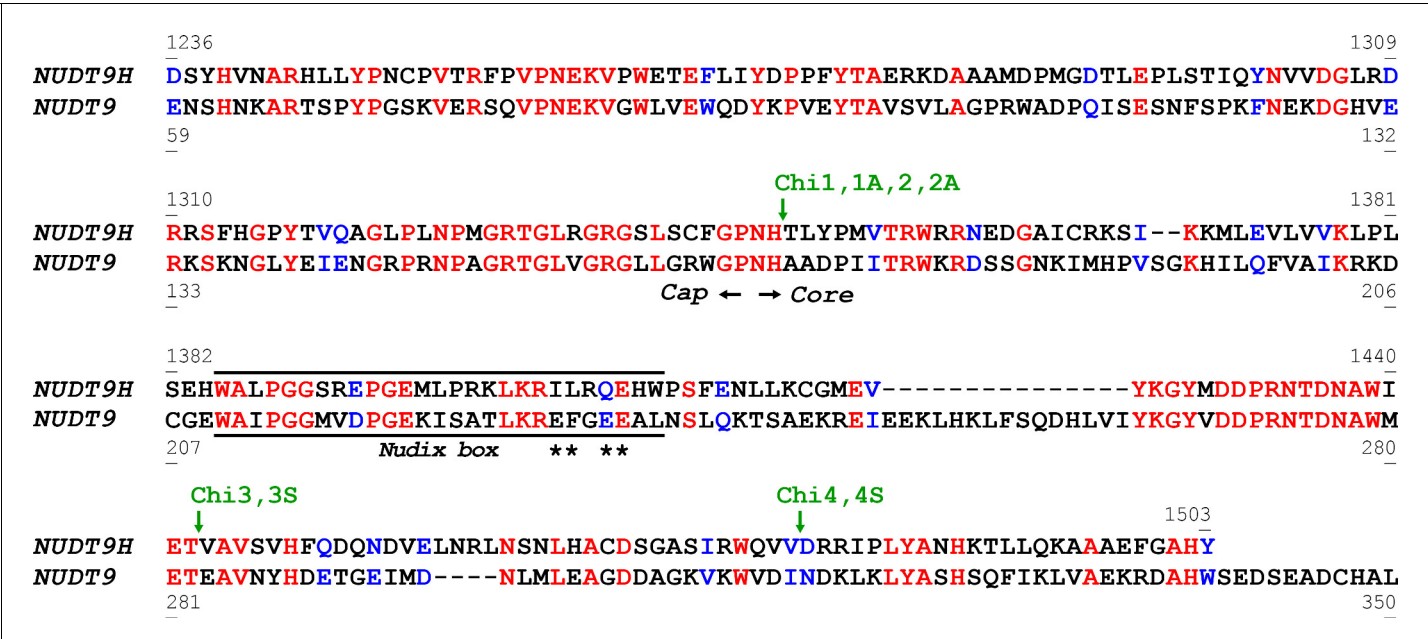

**Figure 2.** Alignment of NUDT9H and NUDT9 sequences. Aligned (*Corpet, 1988*) sequences of NUDT9H and NUDT9, highlighting identical (*red*) and similar (*blue*) residues. The approximate border between Cap and Core subdomains of the proteins is indicated by two *black arrows*. *Black line* identifies the *Nudix* box, *asterisks* denote residues critical for the catalytic activity of NUDT9. *Green arrows* and *labels* indicate transition points for sequence swapping in each chimera (see also *Figure 3*).

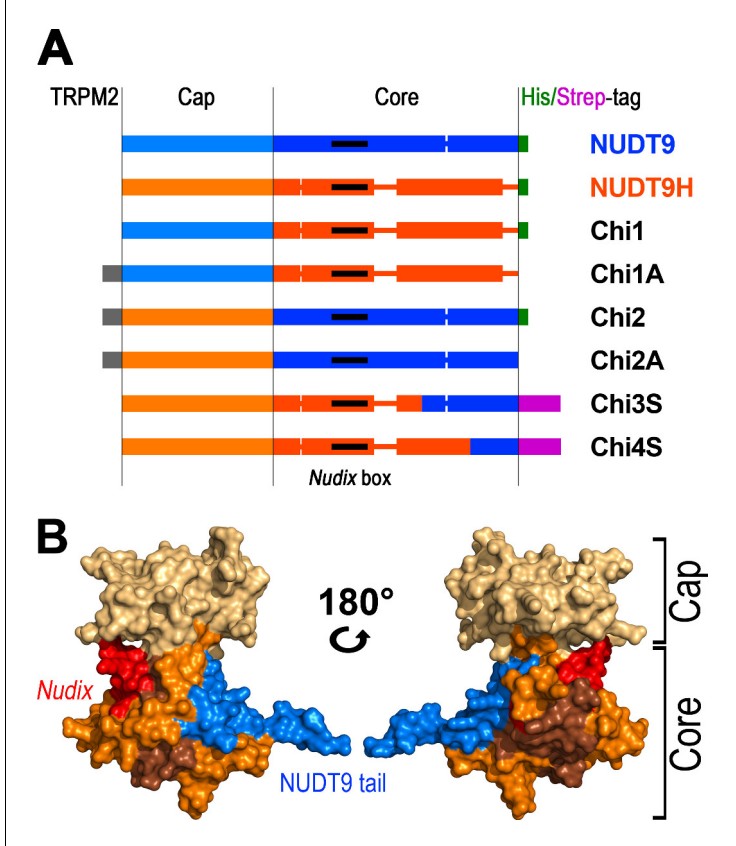

**Figure 3.** Representation of the various recombinant proteins used in this study. (**A**) Aligned block diagrams of recombinant constructs with segment lengths realistically scaled to reflect their actual sizes (numbers of residues). The Cap and Core segments of NUDT9 (*blue*) and NUDT9H (*orange*) are separated by a *vertical black line*. Gaps emerging from the alignment of the two sequences (*Figure 2*) are presented as thick lines with the respective color. The 15 amino-acid stretch from the upstream cytosolic linker sequence of TRPM2 (*gray*), present in some chimeras, contains a native *BlpI*-site which allows subcloning the domain into full-length TRPM2. Some constructs contain a C-terminal hexahistidine tag (*green*) or Twin-Strep tag (*magenta*). *Nudix* boxes within Core segments are indicated as black lines (see also *Table 1*). (**B**) NUDT9 fragments replaced in the chimeras by NUDT9H sequence, mapped onto surface representation of NUDT9 crystal structure (PDB ID: 1Q33). Coloring highlights progressive replacement of NUDT9 sequence with NUDT9H sequence in chimeric constructs. The Cap subdomain (*light brown*) is replaced in Chi2 and Chi2A. A fragment from the Core (*orange*), containing the *Nudix* box (*red*), is also replaced in Chi3S. Additional Core fragment (*dark brown*) further extends the NUDT9H sequence in Chi4S, leaving only 36 C-terminal amino acids from NUDT9 (*blue*).

The following figure supplement is available for figure 3:

**Figure supplement 1.** Purified recombinant proteins.

using a slower induction protocol (0.1 mM IPTG, 25°C, overnight); and replacement of the hexahistidine tag with a Twin-Strep tag (Chi3S and Chi4S) greatly improved protein purity, as confirmed by SDS-PAGE (*Figure 3—figure supplement 1*) and mass spectrometry analysis of the purified Chi4S sample. Indeed, using this slow induction protocol even some soluble NUDT9H protein could be obtained and purified (*Figure 3—figure supplement 1*, right); however, this preparation precipitated overnight at 4°C. Thus, Chi4S remained the best NUDT9H model that we were able to obtain in stable, soluble form (*Table 1*).

**Table 1.** Properties of the recombinant proteins.

| Construct | M.W. (kDa) | Theor. pI* | % NUDT9H[†] | ecOpt[‡] | Nudix[§] | Soluble | Enzyme |
|---|---|---|---|---|---|---|---|
| NUDT9 | 34.0 | 6.88 | 0 | Yes | NUDT9 | Yes | Yes |
| NUDT9H | 31.5 | 8.34 | 100 | Yes | NUDT9H | No[¶] | No |
| Chi1 | 31.7 | 9.30 | 59 | Yes | NUDT9H | No | N/A[††] |
| Chi1A | 32.3 | 9.16 | 59 | No | NUDT9H | No | N/A |
| Chi2 | 35.7 | 6.27 | 41 | No | NUDT9 | Yes | Yes |
| Chi2A | 34.6 | 5.91 | 41 | No | NUDT9 | Yes | N/A |
| Chi3S | 35.4 | 6.40 | 77 | Yes | NUDT9H | Yes (25°C)** | No |
| Chi4S | 35.8 | 7.16 | 91 | Yes | NUDT9H | Yes (25°C)** | No |

\* - Theoretical pI of the construct (*Protein Calculator v3.4*).

† - Percentage of NUDT9H sequence (TRPM2 residues 1236-1503) contained in the construct. (The segment in brackets does not include the 15 amino-acid upstream stretch from TRPM2).

‡ - Gene sequence optimized for expression in *E.coli*. 'No' denotes that the original, human nucleotide sequence (**Nagamine et al., 1998**) was used.

§ - Denotes the origin (NUDT9 or NUDT9H) of the *Nudix* box sequence in the construct.

¶ - Small fraction of the protein is soluble when expressed at 25°C.

** - Small fraction of the protein is soluble when expressed at 37°C, reasonably soluble when expressed at 25°C.

†† - Not Available

### The Nudix box of NUDT9H does not support ADPR hydrolase activity

The solubility and relative stability of mitochondrial NUDT9 and three of our chimeric constructs (Chi2, 3S, 4S) allowed purification of milligram amounts of these recombinant proteins to reasonable homogeneity (*Figure 3—figure supplement 1*) and assessment of their enzymatic activities. Furthermore, despite its long-term instability, we also subjected freshly purified NUDT9H (*Figure 3—figure supplement 1*, right) to enzyme activity assays. Following a 2-hr incubation of 1 µM of each protein with 10 mM ADPR substrate and 16 mM $Mg^{2+}$ (*Perraud et al., 2003*), TLC analysis of the reaction mixtures confirmed that NUDT9 and Chi2 (which contain the NUDT9-type *Nudix* box) are both active ADPRases, with turnover rates sufficient to cause complete – or near-complete – conversion of ADPR into AMP and R5P within this time frame (*Figure 4A and B*; R5P is invisible on the TLC). In contrast, Chi3S, 4S and NUDT9H (which contain the NUDT9H-type *Nudix* box, *Figure 3A*) all failed to hydrolyze the nucleotide under the same conditions (*Figure 4C–E*): this same overall pattern was observed for all six reaction conditions tested, which included three different pH values (6.5, 7.1 and 8.5) in the absence or presence of 100 mM NaCl (*Figure 4A–E*, reaction buffers 1–6).

The reported enzymatic activity of NUDT9H is about 100-fold lower than that of NUDT9 (*Perraud et al., 2001*, *2003*). Closer inspection of the TLC plates of the inactive proteins indeed revealed very weak AMP spots appearing in the reaction mixtures at basic pH (*Figure 4C–E*, lanes 3 and 6). The low sensitivity of the TLC technique requires the presence of at least 0.2–0.3 mM AMP for visual detection (*Figure 4F*, top slice). Thus, assuming 2–300 µM AMP produced by 1 µM protein within our 2-hr incubation would suggest apparent $k_{cat}$ values of 0.03–0.04 $s^{-1}$ for Chi3S, 4S and NUDT9H at pH 8.5 and 25°C, in good agreement with the previous reports on NUDT9H ($k_{cat}$ ~0.06 $s^{-1}$ at pH 9 and 37°C). However, our prior observation of spontaneous degradation of several dinucleotides at basic pH (*Tóth et al., 2015*) prompted us to consider inherent instability of ADPR at basic pH as a possible reason for the observed slow AMP accumulation. Indeed, we found that ADPR undergoes slow degradation to AMP and R5P at pH 8.5 (*Figure 4F*, middle slice; also see *Figure 4—figure supplement 2*), highlighting that care must be taken to distinguish spontaneous hydrolysis from true enzymatic activity. At 25°C and pH 8.5 ~20% of ADPR degraded spontaneously in 24 hr (*Figure 4F*, middle slice, compare the two lanes marked by asterisks), indicating a decay time constant of ~100 h, and thus ~2% degradation in 2 hr: this rate of spontaneous degradation indeed fully accounts for the weak AMP spots observed for Chi3S, 4S, and NUDT9H at basic pH. More importantly, addition of either 0.1 or 1 µM Chi4S to ADPR-containing reaction mixtures at basic pH, either with (RB6) or without (RB3) 100 mM NaCl, did not affect the rate of spontaneous AMP accumulation (*Figure 4F*, bottom slice). This clearly demonstrates that Chi4S does not possess any hydrolytic activity towards ADPR, most

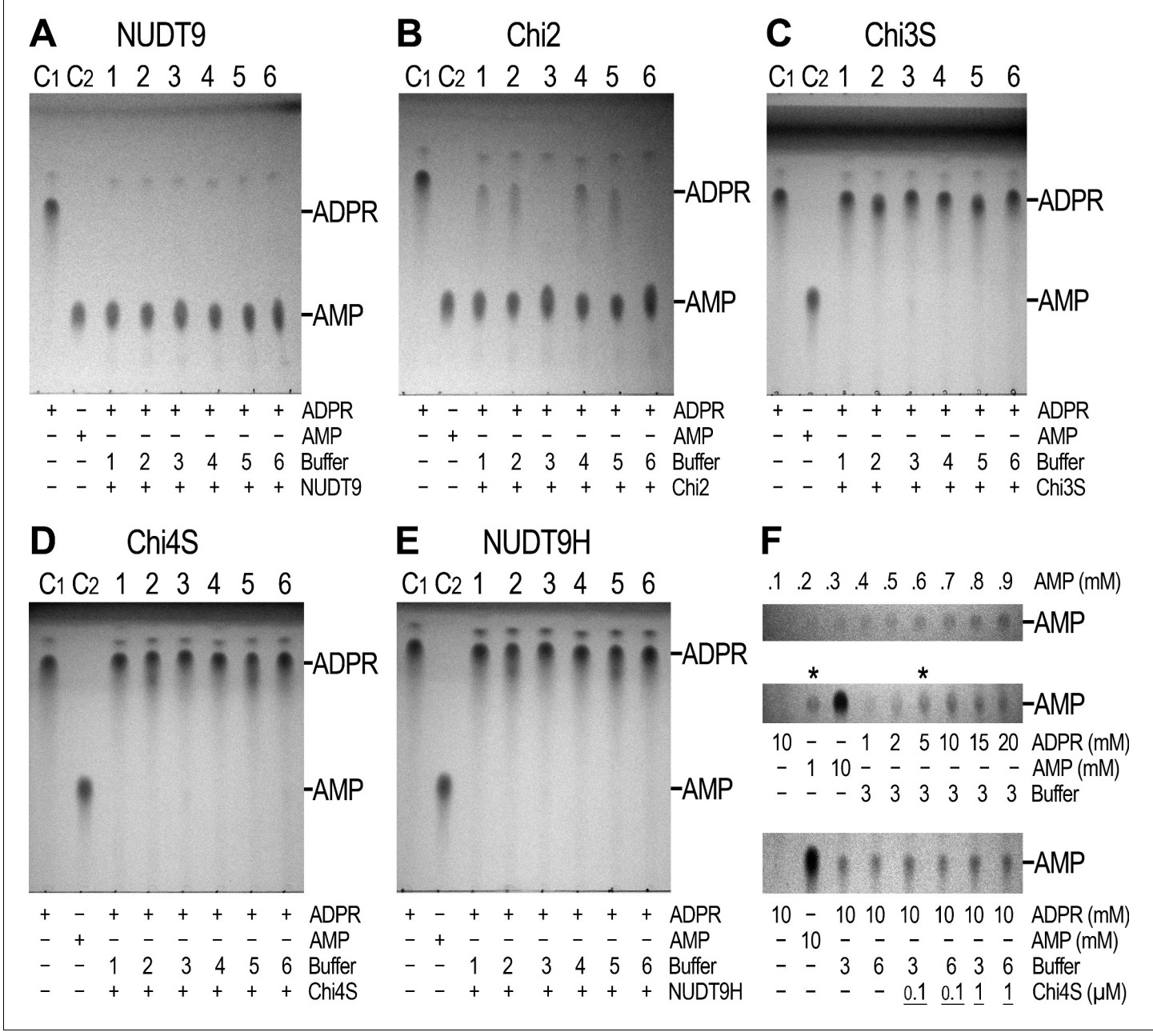

**Figure 4.** Enzymatic activity of NUDT9, NUDT9H, and chimeras. (A–E) 10 mM ADPR (lane C1) and 10 mM AMP (lane C2) controls in water. Lanes 1–6 show reaction mixtures of 1 μM purified NUDT9 (A), Chi2 (B), Chi3S (C), Chi4S (D), or NUDT9H (E), incubated with 10 mM ADPR and 16 mM MgCl$_2$ in 50 mM RB1-6, respectively, for 2 hr at room temperature. (F) Top slice: TLC-detection limit for AMP: 0.1–0.9 mM water-dissolved AMP in 0.1-mM incrementing steps. Middle slice: ADPR self-degradation. The slice shows only the AMP position of (from left to right): 10 mM ADPR control (in water), 1 mM AMP control (in water), 10 mM AMP control (in water), and 1, 2, 5, 10, 15 and 20 mM ADPR incubated for 24 hr at room temperature in RB3 (pH 8.5) (last six lanes). Bottom slice: ADPR self-degradation in the absence and presence of Chi4S, showing only the AMP position. From left to right: 10 mM ADPR control (in water); 10 mM AMP control (in water); 10 mM ADPR in RB3 (pH 8.5); 10 mM ADPR in RB6 (pH 8.5/NaCl); 0.1 μM Chi4S + 10 mM ADPR in RB3; 0.1 μM Chi4S + 10 mM ADPR in RB6; 1 μM Chi4S + 10 mM ADPR in RB3; 1 μM Chi4S + 10 mM ADPR in RB6. Except for the controls in lanes 1–2, all samples also contained 16 mM MgCl$_2$, and were incubated for 24 hr at room temperature. In all Panels 1-μl aliquots were loaded for each sample. The three slices of Panel **F** can be seen as full-scale TLC plates in *Figure 4—figure supplement 1*.

The following figure supplements are available for figure 4:

**Figure supplement 1.** Full-scale TLC images of the slices shown in Panel **F** on *Figure 4*.

*Figure 4 continued on next page*

*Figure 4 continued*

**Figure supplement 2.** Spontaneous ADPR degradation at basic pH, monitored with TLC.

likely due to critical substitutions in its *Nudix* box (*Figure 2*; cf., [*Perraud et al., 2003*]), and that the observed slow accumulation of AMP is merely caused by ADPR instability at basic pH.

## Enzymatic activity of NUDT9 and Chi2 is pH dependent

We noted slower enzymatic activity for Chi2 at pH 6.5 and 7.1 compared to 8.5 (*Figure 4B*), whereas ADPR hydrolysis by NUDT9 appeared complete at all pH values (*Figure 4A*). To test whether such pH dependence might also apply for NUDT9, but could not be observed at the chosen timescale of 2 hr, we analyzed for both enzymes the time course of AMP accumulation in ADPR-saturated reaction mixtures at the three different pH values (in RB1, RB2 and RB3). Both NUDT9 and Chi2 showed a clear preference for basic pH, causing AMP accumulation to be fastest at pH 8.5 and slowest at pH 6.5 (*Figure 5A and B*). The calculated $k_{cat}$ values (in s$^{-1}$) quantify this pH dependence of catalytic rates for both proteins (*Figure 5C*), and reveal higher sensitivity for Chi2 (~5.8-fold decrease from pH 8.5 to 6.5) compared to NUDT9 (~1.4-fold decrease from pH 8.5 to 6.5). Spontaneous degradation of ADPR at basic pH (*Figure 4F*, middle slice) is not a major contributor to this pH dependence, because the total duration of these kinetic experiments (~20 min) was too short for that effect to become significant: correction for the ~30 µM AMP which forms spontaneously in 20 min at pH 8.5 would amount to a downward adjustment of those $k_{cat}$ values (*Figure 5C*, *black bars*) by only ~0.05 s$^{-1}$. Furthermore, whereas we cannot exclude the possibility that the two proteins were applied at slightly different concentrations, and that this may have caused the observed differences in their absolute reaction rates, investigating pH dependence of activity of a given enzyme (i.e. the same protein stock) is not subject to such limitations. Of note, the $k_{cat}$ of 9 s$^{-1}$ for NUDT9 at pH 8.5 is very similar to previously reported values (~8 s$^{-1}$, [*Perraud et al., 2001*, *2003*]).

## High-resolution colorimetric assay confirms enzymatic activity profiles of NUDT9, NUDT9H, and chimeras

The TLC assay allows unambiguous identification of the reaction product AMP based on its chromatographic mobility, but its resolution is limited. As an independent approach, we therefore verified enzymatic activities at both 37°C (*Figure 6A*) and room temperature (RT, *Figure 6B*) for

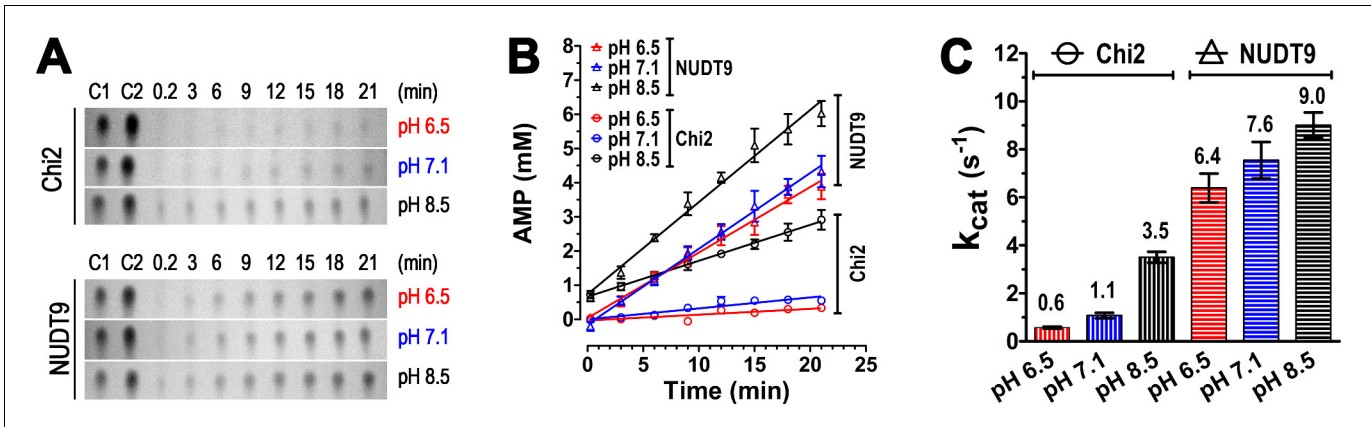

**Figure 5.** Enzyme kinetics of Chi2 and NUDT9. (**A**) Representative TLC images showing the time courses (in minutes) of hydrolysis of ADPR (10 mM) by 0.5 µM Chi2 (*top*) or NUDT9 (*bottom*) at 3 different pH values (RB1, RB2 and RB3). The slices show only the position of the AMP spots. Control 1 (C1): 5 mM AMP in water. Control 2 (C2): 10 mM AMP in water. (**B**) AMP accumulation during ADPR hydrolysis by Chi2 (*circles*) and NUDT9 (*triangles*) at 3 different pH values. Densities of AMP spots (such as those shown in Panel **A**) were normalized to the density of the control AMP spot (5 mM) on the same TLC sheet. For each time point mean ± SEM from 3 experiments is shown. Straight lines are linear regression fits to the averaged data points. (**C**) Estimates of $k_{cat}$ (s$^{-1}$) for both proteins at each pH, calculated from the fitted slopes in B (mean ± SEM; mean values are also printed above each bar).

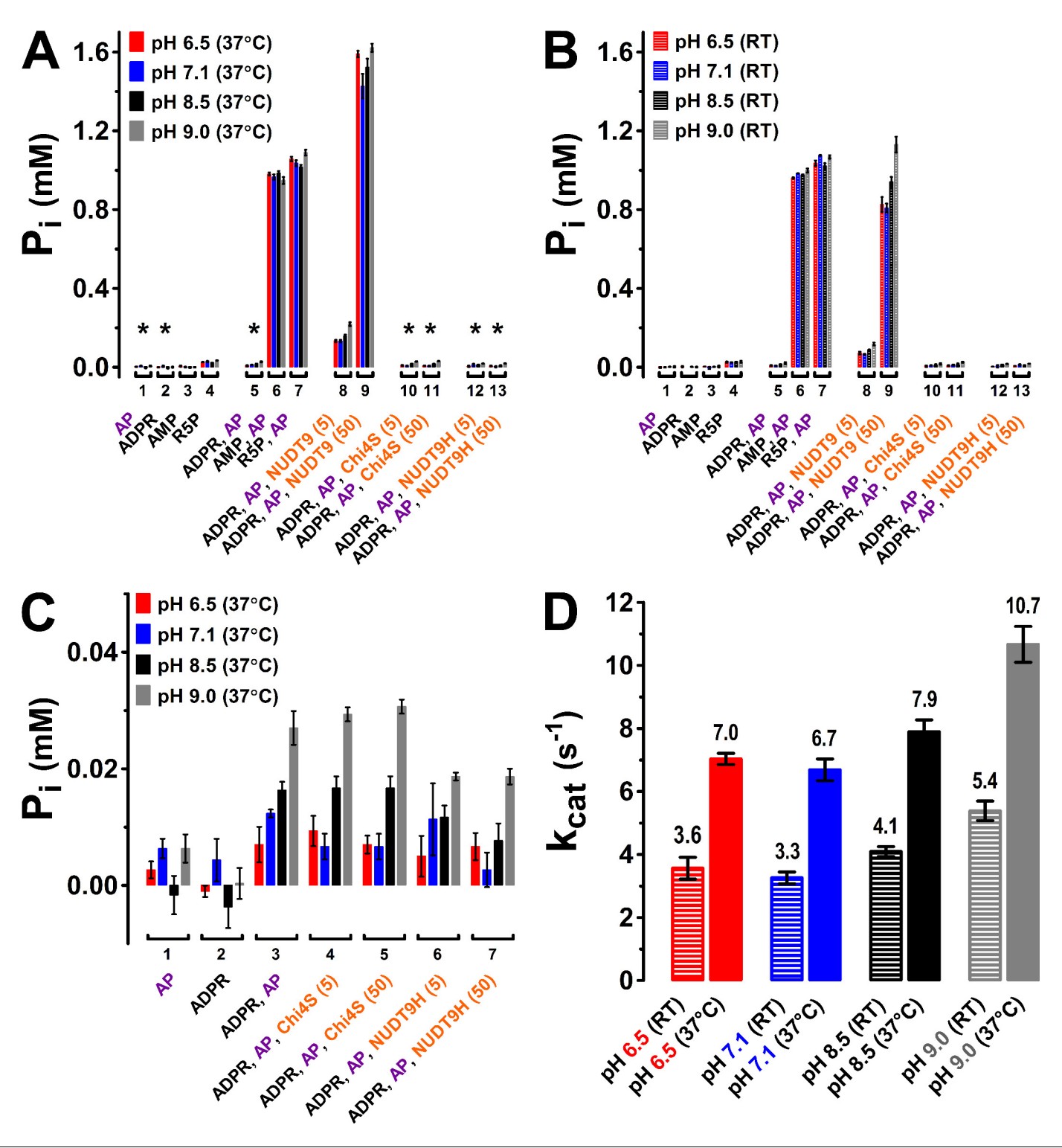

**Figure 6.** Enzymatic activity of NUDT9, Chi4S, and NUDT9H measured with the $P_i$ assay. (**A–B**) $P_i$ (in mM) liberated in sample mixtures containing, as indicated, AP (5–6 U alkaline phosphatase), ADPR (1 mM), AMP (1 mM), R5P (1 mM), and either 5 or 50 nM protein (NUDT9, Chi4S, or NUDT9H), and incubated for 30 min at pH 6.5, 7.1, 8.5, or 9.0 (*color coded*) at either 37°C (**A**) or room temperature (RT) (**B**). *Asterisks* in **A** denote samples replotted with an expanded ordinate in Panel **C**. Bars represent mean ± SEM from 3 independent experiments. (**C**) Expanded display of samples denoted with *asterisks* in Panel **A**. (**D**) Estimated $k_{cat}$ values (in s$^{-1}$) for NUDT9 under each experimental condition (pH and temperature), calculated (see Materials and methods) from [$P_i$] liberated in the samples containing 5 nM NUDT9 (*group 8*) in Panels **A** and **B**.

*Figure 6 continued on next page*

*Figure 6 continued*

The following figure supplements are available for figure 6:

**Figure supplement 1.** Spontaneous ADPR degradation at basic pH, monitored with $P_i$ assay.

**Figure supplement 2.** Size exclusion chromatograms of NUDT9, HUDT9H, and Chi4S.

NUDT9, Chi4S, and NUDT9H using the more sensitive colorimetric detection of inorganic phosphate ($P_i$), liberated from both ADPRase products (AMP and R5P) by co-applied alkaline phosphatase (AP) (*Rafty et al., 2002*). As a validation of this coupled assay, no significant amounts of contaminating $P_i$ were found in either the applied stock of AP (5–6 U), or in 1-mM solutions of ADPR, AMP, or R5P (*Figure 6A–B*, *first four group of bars*). Moreover, in the 30-min time frame of our incubation AP released stoichiometric amounts (~1 mM) of $P_i$ from either AMP or R5P (but not from ADPR), confirming that under our experimental conditions the AP reaction is not rate limiting for the coupled assay (*Figure 6A–B*, *groups 5–7*), releasing 2 mM of $P_i$ per mM ADPR hydrolyzed. When NUDT9 (5 or 50 nM) was co-applied with 1 mM ADPR and AP, the released $P_i$ roughly scaled with the applied amount of protein, signalling specific ADPR hydrolysis by NUDT9 (*Figure 6A–B*, *groups 8–9*). Comparison of released [$P_i$] for assays performed at pH values between 6.5 and 9.0 confirmed modest but significant enhancement of NUDT9 activity at basic pH (*Figure 6A–B*, compare *four colored bars* within *groups 8–9*). Molecular turnover rates ($k_{cat}$), calculated from the 5-nM NUDT9 data, were in the range of ~4–11 s$^{-1}$ (*Figure 6D*), in good agreement with the estimates from the TLC assay (*Figure 5C*). In contrast, co-applying 5 or 50 nM of either Chi4S or NUDT9H with 1 mM ADPR and AP did not result in any significant $P_i$ release (*Figure 6A–B*, *groups 10–13*; expanded in *Figure 6C*): the small concentrations of $P_i$ released in these assays (corresponding to the hydrolysis of ~1% of the applied 1 mM ADPR) clearly reflect spontaneous ADPR hydrolysis (cf., *Figure 6—figure supplement 1*), because they remained insensitive to a ten fold variation in Chi4S or NUDT9H concentration (*Figure 6C*, compare *groups 4–5* and *6–7*), and were not larger than the [$P_i$] released in the absence of any ADPRase (*Figure 6C*, *group 3*).

Importantly, lack of catalytic activity for Chi4S and NUDT9H was not due to microaggregation of these domains in a quasi-stable solution: size exclusion chromatograms for both proteins showed a main monomeric peak and a smaller peak corresponding to a dimer, just as for highly soluble NUDT9 (*Figure 6—figure supplement 2*, compare *red* and *blue* to *black profile*), while higher-stoichiometry aggregates constituted only a minor fraction in each case. For the above functional assays only the pooled fractions corresponding to the monomeric peaks were used.

## Both enzymatically active and inactive chimeras are functional in the context of full-length TRPM2

Enzymatic activity of purified Chi2 confirmed that it is a functional protein, similar to mitochondrial NUDT9 (*Figure 4A–B*). To verify functionality also for enzymatically inactive Chi3(S) and Chi4(S), we tested whether any of these constructs could substitute for the native NUDT9H domain in the context of full-length TRPM2, to confer ADPR-induced gating upon the channel. For this purpose we used the non-inactivating (*Tóth and Csanády, 2012*) T5L-TRPM2 pore mutant (hereafter 'TRPM2'), and subcloned Chi2, Chi3 and Chi4 into TRPM2, replacing its native NUDT9H domain. The three full-length chimeric channels (TRPM2-Chi2, TRPM2-Chi3 and TRPM2-Chi4) were then expressed in *Xenopus laevis* oocytes, and tested for ADPR-dependent channel activation in inside-out patches. For all three constructs cytosolic superfusion with ADPR and Ca$^{2+}$ triggered macroscopic currents (*Figure 7B–D*) resembling those of unmodified T5L-TRPM2 (*Figure 7A*), indicating successful ligand binding to the chimeric domains: in 1 μM ADPR, the previously established half-saturating concentration for TRPM2 (*Tóth and Csanády, 2012*), similar fractional currents were activated for TRPM2, TRPM2-Chi3, and TRPM2-Chi4 (*Figure 7A,C–E*), indicating similar apparent affinities for ADPR. Fractional activation of TRPM2-Chi2 by 1 μM ADPR was reduced (*Figure 7B,E*), reflecting a modest, ~3.5-fold increase in the K$_{1/2}$ for ADPR-induced current activation in TRPM2-Chi2 (*Figure 7H*). The time constants of current decay upon ADPR removal were two–three-fold shorter for the chimeric constructs compared to TRPM2 (*Figure 7F*), whereas the relaxation time constants upon Ca$^{2+}$

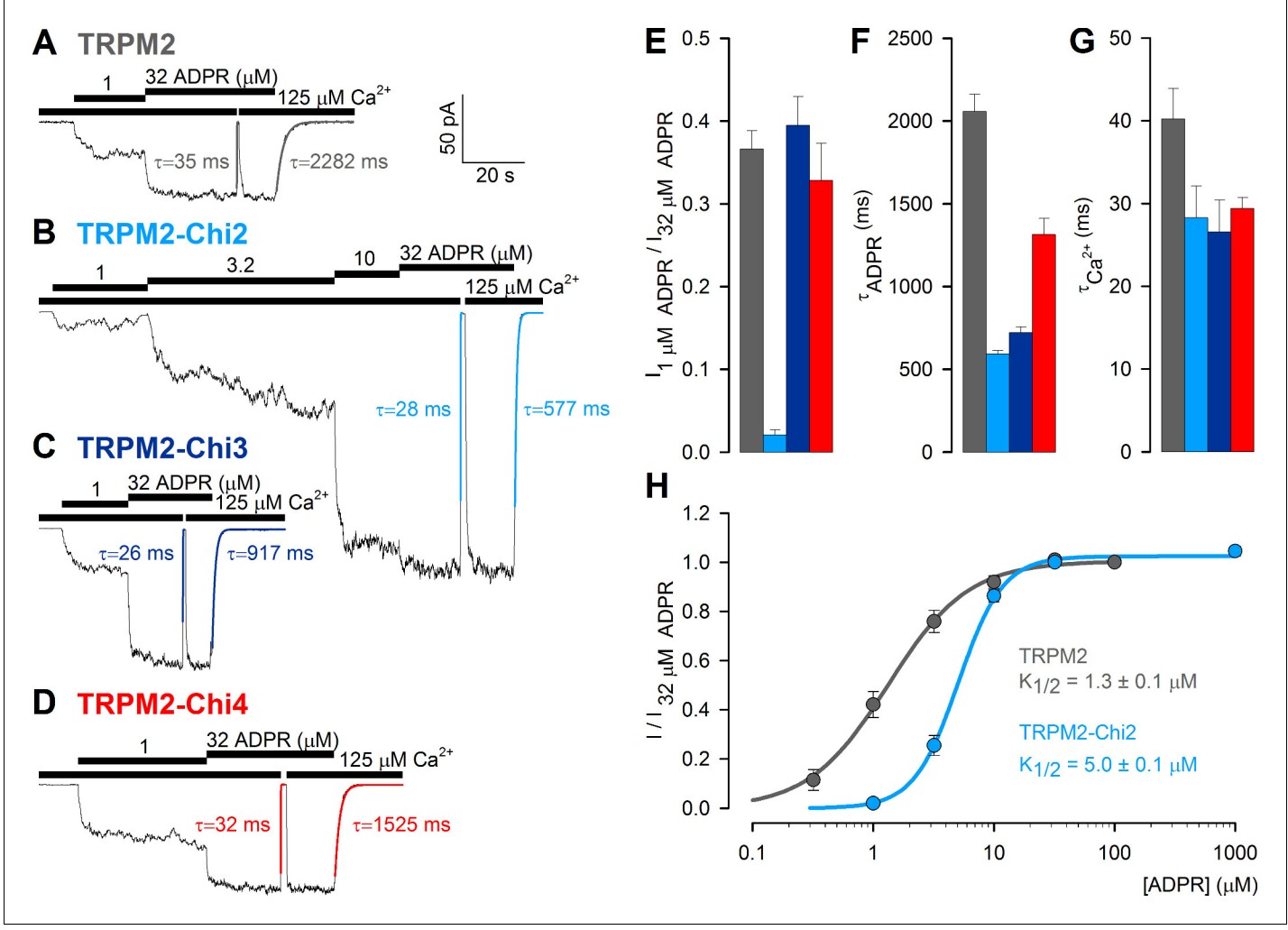

**Figure 7.** ADPR-induced channel activity of full-length TRPM2 chimeras. (A–D) Macroscopic currents in inside-out patches excised from *Xenopus laevis* oocytes expressing (A) T5L-TRPM2, (B) T5L-TRPM2-Chi2, (C) T5L-TRPM2-Chi3, and (D) T5L-TRPM2-Chi4, elicited by superfusion with increasing concentrations of ADPR in the presence of saturating $Ca^{2+}$ (*black bars*). Membrane potential was −20 mV; current relaxation time courses following removal of either $Ca^{2+}$ or ADPR were fitted by single exponentials (*colored lines*) with time constants indicated. (E) Fractional current activation of the constructs in **A–D** by 1 µM ADPR; mean steady currents in 1 µM ADPR were divided by those in 32 µM ADPR in the same patch (mean ± SEM, n≥8). (F–G) Time constants of current relaxation upon sudden removal of ADPR (F) or $Ca^{2+}$ (G) for the constructs in **A–D**, obtained from single-exponential fits (mean ± SEM, n≥5). (H) Dose response curve of fractional current activation for T5L-TRPM2 (*gray*) and T5L-TRPM2-Chi2 (*light blue*); mean steady currents in various test [ADPR] were divided by those in 32 µM ADPR in the same patch. Symbols represent mean ± SEM (n≥5), solid lines are fits to the Hill equation with midpoints printed in the panel.

removal were nearly identical (*Figure 7G*). Overall, apart from moderate kinetic differences, all chimeric channel constructs were gated by ADPR similarly to TRPM2, indicating that their chimeric ligand-binding domains are functional in the context of the full-length channel.

## ADPR analog binds with similar affinities to purified Chi4S protein and full-length TRPM2 channels

Although the electrophysiological experiments confirm functionality for chimeric ligand binding domains in the context of a full-length TRPM2 channel expressed in frog oocytes, we sought to verify functionality also for Chi4S purified from *E. coli*, our best model of isolated NUDT9H. We found that the fluorescent ADPR analog 1, $N^6$-ethenoadenosine-5'-O-diphosphoribose (ε-ADPR) binds to the intact NUDT9H domain of full-length TRPM2, because it suppressed ADPR-evoked TRPM2 currents in a dose-dependent manner: this effect was more pronounced at low, submaximal [ADPR]

(*Figure 8A*), suggesting a competitive inhibitory mechanism. Indeed, the dose response curve for ADPR-activation (*Figure 8B*, *blue symbols and fit line*) was right-shifted in the presence of 100 µM ε-ADPR (*Figure 8B*, *violet symbols*), the fit (*Figure 8B*, *violet fit line*) suggesting a $K_I$ of 112 ± 24 µM for ε-ADPR binding to native NUDT9H.

To test whether binding of ε-ADPR to a soluble protein might be revealed by a change in its fluorescence properties, we first calibrated the dose dependence of ε-ADPR fluorescence in free solution (*Figure 8C*, *black symbols* and *fit line*; see Materials and methods). Applying 100 µM Chi4S caused a clear reduction in ε-ADPR fluorescence (*Figure 8C*, *red symbols*), which reflected fluorescence quenching due to specific binding, because it was not observed upon addition of 100 µM bovine serum albumin (BSA, *gray symbols*). Importantly, Chi4S showed no signs of aggregation even at the high protein concentration used in the assay (*Figure 8—figure supplement 1*) excluding non-specific effects of aggregated protein on fluorescence. Assuming negligible fluorescence for bound ε-ADPR, a safe upper estimate of free [ε-ADPR] could be obtained from the calibration curve (*Figure 8C*, *red projection lines*). A plot of free [ε-ADPR] as a function of total [ε-ADPR] (*Figure 8D*, *red symbols*) was reasonably fitted (*Figure 8D*, *red line*) by a simple binding equation, (see Materials and methods) yielding a $K_d$ (upper) estimate of 111 ± 42 µM for ε-ADPR binding to Chi4S: in good agreement with the $K_I$ of the same analog for native NUDT9H (*Figure 8B*).

## Discussion

The unique group of channel-enzymes (chanzymes), transmembrane proteins that combine ion-channel function with a catalytic activity within a single polypeptide chain, includes the CFTR chloride ion channel, a member of the ATP Binding Cassette (ABC) protein family, and three members of the TRPM channel family, TRPM2, 6, and 7. In the case of CFTR ATP hydrolysis by two cytosolic nucleotide binding domains (NBDs) (*Li et al., 1996*) appears to be an evolutionary vestige inherited from some ancestral ABC exporter (*Gadsby et al., 2006*): whereas in transporters the energy of ATP hydrolysis is harnessed to drive a cycle of conformational changes that result in thermodynamically uphill substrate transport, in CFTR ATP-binding induced dimerization of the two NBDs is strictly coupled to opening, and dimer disruption following ATP hydrolysis to closure, of its transmembrane anion-selective pore (*Csanády et al., 2010*; *Vergani et al., 2005*). Such a nonequilibrium cyclic gating mechanism is unique among ion channels which mediate downhill electrodiffusion of ions, and confers specific properties on CFTR, including stimulation of channel open probability by catalytic-site mutations (*Vergani et al., 2003*), or by drug-induced stabilization of energy barriers (*Csanády and Töröcsik, 2014*). The other well studied chanzyme is the TRPM7 cation channel, which contains a kinase domain fused to its cytosolic C-terminus (*Runnels et al., 2001*; *Nadler et al., 2001*; *Yamaguchi et al., 2001*). In this protein the catalytic activity of the enzymatic domain does not appear to be linked to gating of the channel pore, because phosphotransferase and channel activities can be modulated independently of each other (*Matsushita et al., 2005*; *Schmitz et al., 2003*) and the kinase domain is functional when expressed on its own (*Clark et al., 2008*). Nevertheless, recent work uncovered unexpected important specific functions of TRPM7 enzymatic activity: the kinase domain is proteolytically cleaved and translocates to the cell nucleus where it regulates gene expression by phosphorylating histones in the promoters of TRPM7-dependent genes. Moreover, $Zn^{2+}$ dependent binding of the kinase domain to histone remodeling complexes provides a functional link to the TRPM7 channel pore, a major $Zn^{2+}$ influx pathway (*Krapivinsky et al., 2014*).

In the case of TRPM2 sequence similarity of its C-terminal NUDT9H domain to Nudix-family phosphohydrolases (*Bessman et al., 1996*) played an essential role in the discovery of the channel's activating ligand, ADP ribose (*Perraud et al., 2001*; *Hara et al., 2002*; *Sano et al., 2001*). Early demonstration of ADPR hydrolase activity for the isolated ligand binding domain NUDT9H classified TRPM2 as a chanzyme (*Perraud et al., 2001*, *2003*), and raised intriguing questions about the possible role of this activity: all the more as the reported slow turnover rate (~0.1 s$^{-1}$) falls into the range of channel gating rates (*Tóth and Csanády, 2012*). Of note, while this slower rate of catalysis of NUDT9H, compared to NUDT9, was explained by lack of a conserved glutamate in its Nudix box motif important for $Mg^{2+}$ coordination, the actual catalytic base is highly variable in Nudix enzymes, and has not yet been identified for either of these two proteins (*Shen et al., 2003*). However, when introduced into NUDT9H, catalytic site mutations that abolish, or greatly diminish, ADPRase activity of NUDT9 failed to affect the rates of pore gating, and TRPM2 channels were also readily gated by

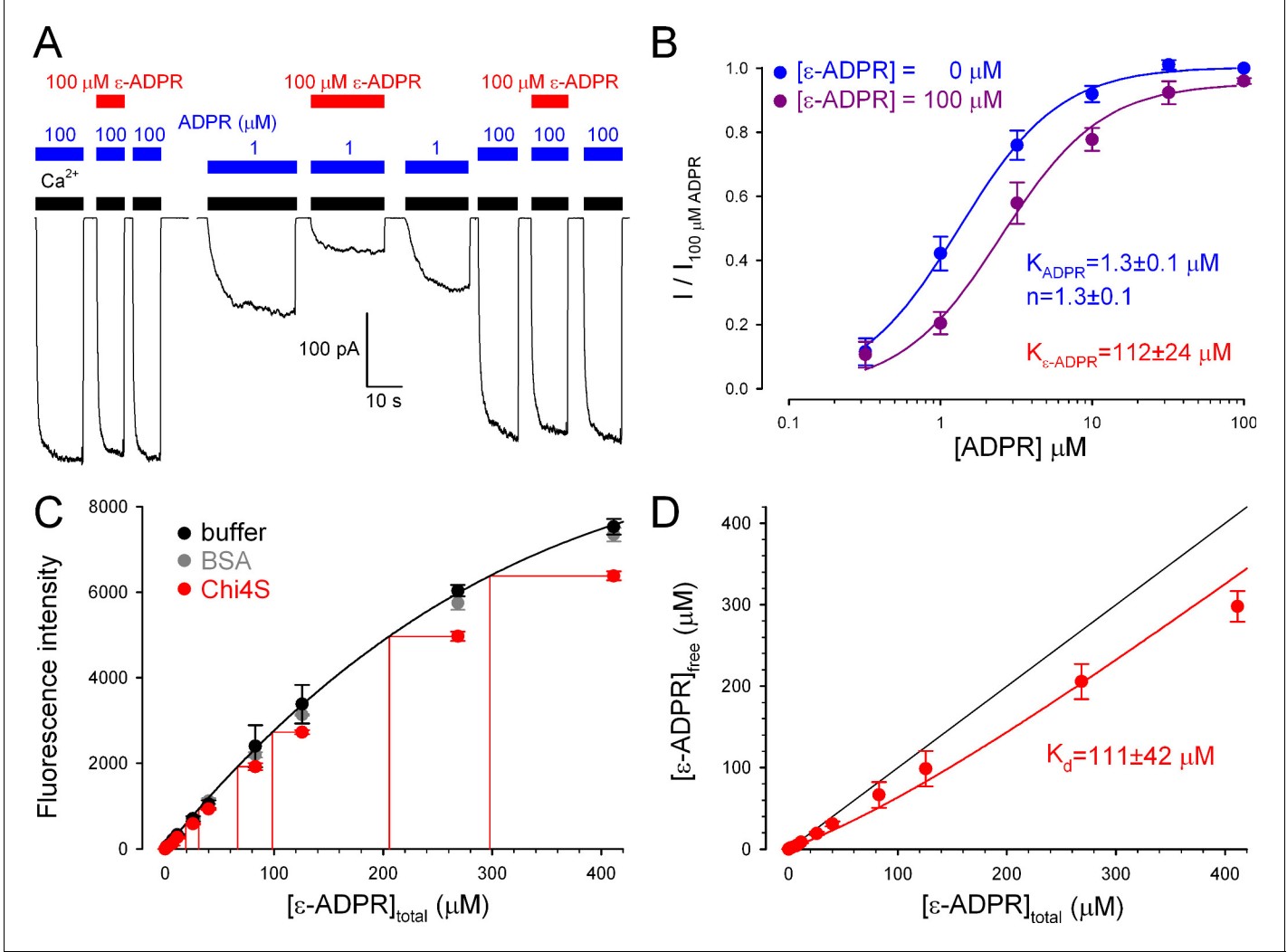

**Figure 8.** Full-length TRPM2 and the isolated Chi4S protein bind ε-ADPR with similar affinities. (**A**) Macroscopic inward T5L-TRPM2 current in inside-out patch excised from *Xenopus laevis* oocyte, elicited at -20 mV by repeated exposures to saturating $Ca^{2+}$ (*black bars*) and 1 or 100 μM ADPR (*blue bars*), in the absence or presence of 100 μM ε-ADPR (*red bars*). (**B**) Dose-dependence of TRPM2 activation by ADPR in the absence (*blue symbols and fit line*) and presence (*violet symbols and fit line*) of 100 μM ε-ADPR (mean ± SEM, n≥7). Printed parameters $K_{ADPR}$, $n$, and $K_{\varepsilon\text{-}ADPR}$ reflect the apparent affinity and Hill slope for ADPR binding, and $K_I$ for ε-ADPR binding, respectively, obtained from the fits (see Materials and methods). (**C**) Fluorescence intensity (arbitrary units, Ex: 310 nm, Em: 420 nm) of ε-ADPR at various total concentrations in the absence of protein (*black symbols*), or in the presence of 100 μM of either BSA (*gray symbols*) or Chi4S (*red symbols*). Symbols plot mean ± SD from four, two, and two independent titrations, respectively, for buffer (*black*), BSA (*gray*), and Chi4S (*red*). *Black line* is a smooth fit of the *black symbols* by a three-parameter empirical mathematical function, and was used as a calibration curve to determine free [ε-ADPR] in the presence of Chi4S, assuming negligible fluorescence of the bound analog (*red projection lines*). (**D**) Plot of upper estimate of free [ε-ADPR] as a function of total [ε-ADPR] in the presence of 100 μM Chi4S (*red symbols*), fitted (*red line*) to a simple binding equation (see Materials and methods) to obtain an upper estimate of $K_d$ (see panel). *Black line* illustrates free [ε-ADPR] expected in the absence of binding.

The following figure supplement is available for figure 8:

**Figure supplement 1.** Lack of effect of five-fold concentration on Chi4S solubility.

a non-hydrolyzable ADPR analog (*Tóth et al., 2014*). These findings ruled out a direct link between the ADPRase cycle and gating of the TRPM2 channel pore itself, but did not eliminate the possibility of some, as yet unidentified, role for this catalytic function. The present work therefore addressed the putative enzymatic activity itself.

To circumvent the problem of low solubility of NUDT9H we sought to establish a chimeric model which resembles the native protein sequence as closely as possible. The region responsible for the low solubility of NUDT9H could be confined to its ~60 C-terminal residues: replacing the entire span (Chi3S), or even just the downstream half (Chi4S) of this C-terminal segment with homologous NUDT9 sequence (*Figure 3*) was sufficient to confer solubility upon NUDT9H. The C-terminus of NUDT9 is surface exposed ([*Shen et al., 2003*]; PDB ID: 1Q33; 1QVJ), free to interact with other protein molecule(s), and presumably the same is true for the C-termini of the chimeras and NUDT9H. Therefore, among other reasons, the cumulative charge of the C-terminus may play an important role in determining their aggregation propensity during expression in the bacterial cytoplasm (at around neutral pH). The theoretical total charges of the ~60–70 C-terminal residues of our constructs (ignoring affinity tags) are −8.1 (NUDT9, Chi2 and Chi3), +0.9 (NUDT9H), −2.9 (Chi4) at pH 7.0. It is thus possible that the relative charge-neutrality of the C-terminus of NUDT9H contributes to its insolubility.

Such low solubility, when expressed in isolation, for a cytoplasmic protein domain like NUDT9H – in striking contrast to its very soluble mitochondrial homologue – is somewhat unexpected. A likely explanation is that NUDT9H natively exists in the context of tetrameric TRPM2, and must share an interface with other channel regions which transduces ligand binding into pore opening. Whether NUDT9H interacts with the large N-terminal domain of the channel, with the cytosolic loops connecting the transmembrane helices, and/or with other NUDT9H domains remains to be elucidated, but it is tempting to speculate that the C-terminal ~20% of NUDT9H sequence might be directly involved in such interactions. Consistent with such an interpretation, incremental replacement of NUDT9H segments with corresponding NUDT9 sequences (TRPM2(-NUDT9H) → TRPM2-Chi4 → TRPM2-Chi3 → TRPM2-Chi2) causes a parallel graded decrease in the stability of the open-channel state, as reflected by a speeding of channel closure upon ADPR removal (*Figure 7F*; *gray → red → dark blue → light blue bars*).

NUDT9, the soluble chimeric constructs generated by sequence swapping, and even some NUDT9H could be purified and tested for enzymatic activity. These experiments revealed a distinct pattern directly correlated with the origin of the *Nudix* box in each protein: constructs possessing the *Nudix* box of NUDT9 (NUDT9 and Chi2) were active ADPR hydrolases, whereas those containing the *Nudix* box of NUDT9H (Chi3S, Chi4S and NUDT9H) were inactive (*Figures 4* and *6*). Notably, our soluble chimeric model most similar to the native NUDT9H sequence, Chi4S, is ~90% identical to NUDT9H (including the *Nudix* box region) and clearly capable of ligand binding (*Figures 7D*,*8C–D*), but did not exhibit any sign of enzymatic activity under the various reaction conditions tested.

Previous studies reported a weak enzymatic activity for the isolated NUDT9H domain of TRPM2 (*Perraud et al., 2001*, *2003*). While we did also detect slow accumulation of AMP when Chi3S, Chi4S, or NUDT9H was incubated with ADPR under alkaline conditions, we fully attribute those results to inherent instability of ADPR at basic pH. Indeed, we observed gradual accumulation of AMP even in protein-free samples of ADPR incubated in alkaline buffers (*Figures 4F*,*6C*, *Figure 4—figure supplement 2*, and *Figure 6—figure supplement 1*) and, assuming an exponential decay time course, estimated a time constant of ~100 hr for spontaneous ADPR hydrolysis at 25°C and pH 8.5. Spontaneous degradation of ADPR at basic pH (~0.3% and ~2% within 20 min and 2 hr, respectively, at 25°C and pH 8.5; ~1% within 30 min at 37°C and pH 9 [*Figure 6—figure supplement 1*]) did not significantly distort the results of the 20- to 30-min enzyme kinetics measurements for the active constructs (*Figures 5C* and *6D*), but fully accounted for the weak AMP signals for our inactive constructs observed after 2-hr incubations using TLC (*Figure 4C–E*, *lanes 3 and 6*), or already after 30-min incubations using the more sensitive $P_i$ detection assay (*Figure 6A–C*): this phenomenon fully explains the reported weak activity of NUDT9H at 37°C and pH 9 (*Perraud et al., 2001*, *2003*). Of note, spontaneous degradation affects enzymatic ADPR hydrolysis assays only at basic, but not at neutral or slightly acidic pH at which ADPR is inherently stable (*Figure 4C–E*, lanes 1, 2, 4, 5; *Figure 4—figure supplement 2*, and *Figure 6—figure supplement 1*).

The catalytic properties of our two enzymatically active recombinant proteins (NUDT9 and Chi2) were largely similar. NUDT9 appeared slightly faster than Chi2, which may be attributed either to slight structural mismatch between 'foreign' Cap and Core segments in Chi2, or to errors in estimating their concentrations. Both enzymes exhibited a modest but clear pH-dependence, with a preference towards basic pH (*Figure 5C*, and *Figure 6D*). Perhaps NUDT9, which natively resides in

mitochondria, might have adapted to this environment, and Chi2, which contains ~60% of the NUDT9 sequence (the entire Core subdomain), presumably shares most of its preferences. In contrast, the cytosolic NUDT9H domain of plasma-membrane resident TRPM2 faces the neutral pH of the cytosol. For this reason, care was taken to test the NUDT9H-like proteins (Chi3S, Chi4S and NUDT9H) in buffers at different pH values (*Figures 4* and *6*).

When introduced into full-length TRPM2 in place of the native NUDT9H domain, chimeric constructs Chi2, Chi3 and Chi4 supported ADPR-induced channel activity which resembled that of intact TRPM2 channels, except for a somewhat reduced apparent affinity of TRPM2-Chi2 (*Figure 7*). One possible reason for this small defect is that the other constructs contain catalytically inactive *Nudix* domains, whereas ADPR hydrolysis by Chi2 might reduce fractional saturation of this domain at low (~1 μM) ADPR concentrations. Alternatively, the significantly larger portion of non-native (i.e. NUDT9-like) sequence in Chi2 (the entire Core subdomain) might compromise formation of the proper interface responsible for transducing ligand binding into channel opening. Nevertheless, all full-length chimeric TRPM2 constructs formed functional channels, demonstrating that NUDT9/NUDT9H chimeric domains are properly folded and competent to bind ADPR and gate the channel. Importantly, TRPM2-Chi4 is almost identical in sequence to the background T5L-TRPM2, its channel behaviour is almost indistinguishable from that of T5L-TRPM2, and we have clearly shown that its isolated ligand-binding domain (Chi4S) – although capable of binding an ADPR analog with a similar affinity as the intact NUDT9H domain of a full-length TRPM2 channel (*Figure 8*) – is not catalytically active. Based on these results, we conclude that the NUDT9H domain of TRPM2 is not an ADPR hydrolase, and that TRPM2 is a simple ligand-gated channel, not a 'chanzyme', despite the fact that its ligand binding domain is clearly a member of the family of Nudix enzymes. One potential advantage of its broken enzymatic activity might be increased ligand occupancy at low ADPR concentrations. The Chi4S construct developed here is a convenient soluble model of NUDT9H which might be useful for future structural and biochemical studies.

## Materials and methods

### Molecular biology

Except for Chimera 2 (Chi2), all recombinant proteins for bacterial expression were synthesized and incorporated into the pJ411 plasmid which bears kanamycin resistance (DNA2.0, Menlo Park, CA, USA). Chi2 was obtained by PCR-amplification of Chi2A cDNA using forward primer (5′-TTTAAC TTTTAGGAGATAAA<u>CATATG</u>CCGGATGCTGAG, *NdeI* site underlined) and reverse primer (5′-AACGTCTG<u>CTCGAG</u>CAACGCATGGCAGTCAGCTTCAGA, *XhoI* site underlined), and subcloned into the pET21c vector. NUDT9, NUDT9H, Chi1 and Chi2 had C-terminal hexahistidine tags, while Chi3S and Chi4S had C-terminal Twin-Strep tags. The N-termini of Chi1A, Chi2, Chi2A, Chi3, and Chi4 included an upstream stretch of 15 TRPM2 residues which contained a native *BlpI*-site (*Figure 3A*, *gray*). The pJ411 plasmids harboring the individual constructs were transfected into inducible *E.coli*, strain BL21 (DE3).

The noninactivating TRPM5-like pore mutant of TRPM2 (T5L-TRPM2) was contained in the pGEMHE vector (*Tóth and Csanády, 2012*). The NUDT9H domain of TRPM2 was replaced by Chi2A, Chi3 or Chi4 using the native *BlpI*-site upstream of the domain and an *XbaI*-site downstream of the Stop-codon in each chimeric construct. Single-base mutations to generate Stop-codons immediately upstream of the hexahistidine tags of Chi3 and Chi4 were then introduced (QuickChange II, Agilent Technologies, Santa Clara, CA, USA), yielding TRPM2-Chi3 and TRPM2-Chi4; TRPM2-Chi2 lacked a His-tag because it was derived from Chi2A.

All constructs were sequence-verified. The TRPM2/pGEMHE plasmids were linearized with *NheI* (New England Biolabs, Ipswich, MA, USA) and transcribed in vitro using T7 polymerase; cRNA was stored at −80°C.

### Protein expression and purification

The soluble proteins (NUDT9 and Chi2) were expressed and purified as previously described for mitochondrial NUDT9 (*Tóth et al., 2014*). In brief, bacterial cultures were grown at 37°C in Luria-Bertani medium supplemented with the appropriate antibiotic and induced with 1 mM isopropyl-β-D-1-thiogalactopyranoside (IPTG) upon reaching $OD_{600}$ of ~0.7. After 3 hr of incubation, cells were harvested

and lysed with sonication in 20 mM Tris (pH 8.5 with HCl) / 200 mM NaCl. The cleared supernatant was incubated with Ni-Sepharose 6 Fast Flow resin (GE Healthcare Hungary, Hungary) for 3 hr under gentle rotation. The resin was then washed with lysis buffer supplemented with 20 mM imidazole, and the protein eluted with a buffer containing 20 mM Tris (pH 8.5 with HCl), 100 mM NaCl, and 400 mM imidazole.

Chi3S, Chi4S and NUDT9H were little soluble to insoluble when expressed at 37°C, the majority of the proteins ending up in inclusion bodies, but gave tractable (~1–5%) soluble fractions when induced with 0.1 mM IPTG at 25°C overnight. NUDT9H was then purified as NUDT9. Chi3S and Chi4S were purified using Strep-Tactin chromatography following the manufacturer's instructions (IBA GmbH, Göttingen, Germany).

Following affinity chromatography, all proteins were subjected to a gel filtration purification step in 20 mM Tris (pH 8.5), 100 mM NaCl (Superdex 200 10/300 GL, GE Healthcare Hungary). Affinity tags were not removed. Protein concentrations were determined spectrophotometrically using theoretical molar extinction coefficients ($\varepsilon_o \sim 55{,}000$ $M^{-1}cm^{-1}$ at $\lambda = 280$ nm). Protein purity was visually checked with SDS PAGE. Protein identity was confirmed by the band position, the main peak position on the gel filtration profile, and, in the case of Chi4S, also by mass spectrometry (Proteome Factory AG, Berlin, Germany). NUDT9 and Chi2 were flash-frozen in liquid nitrogen and kept at −20°C until used. Chi3S, Chi4S and NUDT9H were kept at 4°C, and immediately used for experiments.

## Enzymatic activity assay using thin-layer chromatography (TLC)

ADPRase activities of mitochondrial NUDT9, NUDT9H, and the purified chimeric constructs (Chi2, Chi3S, Chi4S) were quantitated by densitometric detection of the AMP product on TLC sheets. The 20-μl reaction mixtures contained 0.1, 0.5 or 1 μM purified protein, 10 mM Na-ADPR (Sigma-Aldrich, St. Louis, MO, USA), 16 mM $MgCl_2$, and 50 mM reaction buffer (RB). The reaction buffers used were: 50 mM MES, pH 6.5 with NaOH (RB1); 50 mM HEPES, pH 7.1 with NaOH (RB2); 50 mM Tris, pH 8.5 with HCl (RB3); RB1 + 100 mM NaCl (RB4); RB2 + 100 mM NaCl (RB5), and RB3 + 100 mM NaCl (RB6). For the enzyme kinetics experiments on NUDT9 and Chi2, aliquots from the reaction mixtures were collected at the indicated time points; 3 independent experiments were performed for both proteins and each condition, and data are presented as mean ± SEM. For other experiments, the total incubation time was as indicated in the figures legends. All incubations were at room temperature (~25°C). For TLC, 1-μl aliquots of reaction mixtures were placed on Polygram SIL G/UV254 plates (Macherey-Nagel, Düren, Germany), dried and developed in 0.2 M $NH_4HCO_3$ in ethanol:water 7:3 (vol/vol). Nucleotides were visualized under UV light. Densitometry of nucleotide intensities was done using ImageJ software.

## Enzymatic activity assay using colorimetric detection of inorganic phosphate ($P_i$)

The ADPRase activities of the same proteins were additionally tested with a more sensitive assay based on colorimetric detection of $P_i$ ($P_i$ assay) liberated from both ADPRase products AMP and R5P by co-applied alkaline phosphatase (AP) (*Rafty et al., 2002*). AP liberates $P_i$ from AMP and R5P, but not from intact ADPR: thus, $P_i$ is produced from hydrolyzed ADPR at a molar ratio of 1:2. In brief, 150-μl reaction volumes containing 5 or 50 nM purified protein, 5–6 U bovine AP, 1 mM Na-ADPR, 16 mM $MgCl_2$, and 50 mM reaction buffer at various pH values (MES 6.5, HEPES 7.1, Tris 8.5, and Tris 9.0) were incubated for 30 min at either 37°C or room temperature. The reaction was stopped and $P_i$ visualized by adding 850 μl coloring solution (6:1 vol/vol ratio mixture of 0.42% ammonium molybdate tetrahydrate in 1N $H_2SO_4$ and 10% L-ascorbic acid) followed by incubation for 20 min at 45°C. Optical density was then measured at 820 nm (NanoPhotometer P300, Implen GmbH, Munich, Germany). Coloring solution was freshly made, and standard curve (0.050–2 mM $KH_2PO_4$) obtained, daily. Reactions with 1 mM AMP or R5P (instead of ADPR) served as positive controls. All reagents were from Sigma-Aldrich. Assays were performed in triplicates and data are displayed as mean ± SEM.

Molecular turnover rates ($k_{cat}$) for NUDT9 were calculated from the [$P_i$] released in samples containing 5 nM NUDT9, in which [ADPR] remained >0.9 mM (i.e., saturating [*Perraud et al., 2003*]) throughout the 30-min incubation period, using the equation $k_{cat} = 0.5[P_i]_{released}/5$ nM/1800 s.

## Ligand binding assay based on quenching of ε-ADPR fluorescence

The ligand binding ability of purified Chi4S protein was assessed using fluorescence quenching of the ADPR analogue 1, $N^6$-ethenoadenosine-5'-O-diphosphoribose (ε-ADPR - BIOLOG Life Science Institute, Bremen, Germany). A 350-µl volume of 100 µM Chi4S protein in 20 mM Tris (pH 8.5)/ 100 mM NaCl buffer, supplemented with 16 mM $MgCl_2$, was titrated by increasing concentrations of ε-ADPR. Fluorescence emission scans (350–700 nm) were recorded at a fixed excitation wavelength of 310 nm (Hitachi F-7000, Hitachi High Technologies, Maidenhead, UK). Fluorescence emission for ε-ADPR peaks at ~410 nm. Protein-free buffer or 100 µM bovine serum albumin (BSA; instead of Chi4S) were used as controls. Data are shown as mean ± S.D. of two independent experiments.

Dose dependence of fluorescence (Em = 420 nm) of free ε-ADPR (in buffer) was fitted by an empirical non-linear function to produce a smooth calibration curve, used to obtain upper estimates of free [ε-ADPR] in the presence of 100 µM Chi4S protein and various total [ε-ADPR], assuming negligible fluorescence for the bound, relative to the free, form of the analog. The plot of free vs. total [ε-ADPR] was fitted by the simple binding equation

$$[\varepsilon\text{-ADPR}]_{\text{free}} = 0.5\left([\varepsilon\text{-ADPR}]_{\text{total}} - [P]_{\text{total}} - K_d + \sqrt{\left([\varepsilon\text{-ADPR}]_{\text{total}} - [P]_{\text{total}} - K_d\right)^2 + 4K_d[\varepsilon\text{-ADPR}]_{\text{total}}}\right),$$

where $[P]_{\text{total}}$ is the total concentration of Chi4S protein (100 µM), and $K_d$ is a free parameter. Given that the values of free [ε-ADPR] are upper estimates, this approach provides an upper estimate of the $K_d$.

Because for the binding assay the Chi4S preparation had to be concentrated by ~five-fold to reach ~100 µM (3.6 mg/ml), the effect of such a concentration step on Chi4S solubility was verified by subjecting five-fold concentrated Chi4S protein to a second round of size exclusion chromatography. The elution profile of this second run (*Figure 8—figure supplement 1*, *dashed blue profile*) resembled that of the first run, showing a main monomeric peak with a smaller peak corresponding to a dimer. Importantly, larger aggregates did not reappear. This confirms that the signal observed in the fluorescent binding assay is not due to non-specific effects of aggregated protein.

## Isolation and injection of *Xenopus* oocytes

Oocytes isolated from anaesthetized *Xenopus laevis* were digested with collagenase, injected with 10–70 ng cRNA, and stored at 18°C. Recordings were done 2–3 days after injection.

## Excised inside-out patch-clamp recordings

The tip of the patch pipette was filled to ~1 cm height with 140 mM Na-gluconate, 2 mM Mg-gluconate$_2$, 10 mM HEPES (pH 7.4 with NaOH); free $[Ca^{2+}]$ ~4 µM. The pipette electrode was placed into a 140-mM NaCl-based solution carefully layered on top. Bath solution contained 140 mM Na-gluconate, 2 mM Mg-gluconate$_2$, 10 mM HEPES (pH 7.1 with NaOH), and either 1 mM EGTA (to obtain 'zero' (~8 nM) $Ca^{2+}$) or 1 mM Ca-gluconate$_2$ (to obtain 125 µM free $Ca^{2+}$) (*Csanády and Törocsik, 2009*). Na-ADPR was used at final concentrations of 0.32, 1, 3.2, 10, 32, 100, or 1000 µM, and the Na$^+$ salt of ε-ADPR at 100 µM. Macroscopic TRPM2 currents were recorded at 25°C at a membrane potential of −20 mV. Currents were amplified, filtered at 2 kHz (Axopatch 200B), digitized at 10 kHz (Digidata 1322A), and stored to disk (Pclamp9, Molecular Devices, Sunnyvale, CA, USA).

Dose dependence of ADPR-induced current activation, expressed as the ratio of mean macroscopic currents observed in the presence of various test [ADPR] ($I$) and 32 (or 100) µM ADPR ($I_{\text{max}}$) in the same patch, was fitted to the Hill equation ($I/I_{\text{max}} = [ADPR]^n/(K^n + [ADPR]^n)$). Dose dependence of ADPR-activation in the presence of 100 µM ε-ADPR was expressed as the ratio of the currents observed in the presence of various test [ADPR] + 100 µM ε-ADPR ($I$) and 100 µM ADPR alone ($I_{\text{max}}$) in the same patch, and was fitted to the Hill equation substituting $K' = K(1+[\varepsilon\text{-ADPR}]/K_I)$ instead of $K$ ($K$ and $n$ fixed to their control values). Macroscopic current decay time courses following removal of ADPR or $Ca^{2+}$ were fitted by single exponentials using the method of least squares.

## Statistics

Data are displayed as mean ± SEM from $n$ independent experiments, as specified in each figure legend. All data were included in the analysis. Enzyme activities were assayed in at least three

independent experiments. Gating parameters for full length TRPM2 and chimeric channels were estimated from current recordings obtained from at least five patches in each case. These numbers of technical replications allowed us to limit SEM to $\leq 20\%$ of the mean for each measured parameter.

## Acknowledgements

We thank Dr. Christos Chinopoulos for the use of the Hitachi F-7000 fluorometer. Supported by an International Early Career Scientist grant from the Howard Hughes Medical Institute, and MTA Lendület grant LP2012-39/2012 to LC.

## Additional information

### Funding

| Funder | Grant reference number | Author |
|---|---|---|
| Howard Hughes Medical Institute | International Early Career Scientist grant, 55007416 | László Csanády |
| Magyar Tudományos Akadémia | Lendület grant, LP2012-39/2012 | László Csanády |

The funders had no role in study design, data collection and interpretation, or the decision to submit the work for publication.

### Author contributions

II, Conception and design, Acquisition of data, Analysis and interpretation of data, Drafting or revising the article; CM, BT, Acquisition of data, Analysis and interpretation of data; LC, Conception and design, Analysis and interpretation of data, Drafting or revising the article

### Author ORCIDs

Iordan Iordanov, http://orcid.org/0000-0001-8251-5857
Csaba Mihályi, http://orcid.org/0000-0001-7536-3066
László Csanády, http://orcid.org/0000-0002-6547-5889

### Ethics

Animal experimentation: This study was performed in strict accordance with the recommendations in the Guide for the Care and Use of Laboratory Animals of the National Institutes of Health. All of the animals were handled according to approved institutional animal care and use committee (IACUC) protocols of Semmelweis University (22.1/1935/3/2011).

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
