## [Decision Letter]

Thank you for submitting your article "The proposed channel-enzyme Transient Receptor Potential Melastatin 2 does not possess ADP ribose hydrolase activity" for consideration by *eLife*. Your article has been favorably evaluated by Richard Aldrich (Senior Editor) and three reviewers, one of whom, David E Clapham (Reviewer #1), is a member of our Board of Reviewing Editors, and another one is Merritt Maduke (Reviewer #3).

The reviewers have discussed the reviews with one another and the Reviewing Editor has drafted this decision to help you prepare a revised submission.

Summary:

Transient Receptor Potential Melastatin 2 (TRPM2) is a Ca-permeable cation channel activated by ADP ribose(ADPR) binding to its C-terminal cytosolic NUDT9-homology (NUDT9H) domain. This domain is homologous to the soluble mitochondrial ADPR pyrophosphatase (ADPRase) NUDT9, but as shown here, does not hydrolyze ADPR. Reported ADPR hydrolysis classified TRPM2 as a channel-enzyme, but insolubility of isolated NUDT9H hampered further study. The authors develop a soluble NUDT9H, by replacing incrementally extended N-terminal portions of NUDT9 with homologous NUDT9H sequence. When expressed in *E. coli*, chimeras containing up to ~90% NUDT9H sequence remained soluble and were affinity-purified. In ADPRase assays, the conserved Nudix-box sequence of NUDT9 proved essential for activity (*k*_cat_~4-9s^-1^) – NUDT9H itself did not support catalysis. Replacing NUDT9H in full-length TRPM2 with soluble chimeras retained ADPR-dependent channel gating, confirming functionality of chimeric domains. Based on these results, the authors conclude that the NUDT9H domain of TRPM2 is not an ADPR hydrolase, and that TRPM2 is a simple ligand-gated channel, not a "chanzyme". The authors speculate that the potential advantage of its broken enzymatic activity might be increased ligand occupancy at low ADPR concentrations.

This is excellent work, well documented and controlled. This is an outstanding example of careful science well-executed.

Essential revisions:

Overall, the experimental results are beautifully presented. However, the conclusion that the purified domains have no activity remains undemonstrated because it has not been shown that these domains are truly soluble and monodispersed. With expression at 37°C, the domains are completely insoluble; with expression at 25°C, only a tiny fraction of the protein is obtained in the soluble fraction. So, one wonders whether the domains are only quasi-stable and prone to micro-aggregation in solution. The authors perform size exclusion chromatography as a purification step, but none of the chromatograms are shown. Do the domains run as monomers or oligomers? Are there significant aggregate peaks in the chromatograms?

Are the domains monodispersed when re-run on size exclusion chromatography, or are they only quasi-stable? Do the proteins remain monodispersed after concentration for the binding assays? These questions need to be addressed in order for the conclusion that there is no catalytic conclusion to be certain. While the binding assay in principle could confirm that the domains are well behaved, the weak signal observed in the fluorescent binding assay could be due to non-specific effects of aggregated protein on fluorescence rather than to specific binding. Since BSA is a highly soluble protein, it is not a great control in this case. In summary, the chromatograms should be shown and these points discussed.

---

## [Author Response]

*Essential revisions:*

*Overall, the experimental results are beautifully presented. However, the conclusion that the purified domains have no activity remains undemonstrated because it has not been shown that these domains are truly soluble and monodispersed. With expression at 37°C, the domains are completely insoluble; with expression at 25°C, only a tiny fraction of the protein is obtained in the soluble fraction. So, one wonders whether the domains are only quasi-stable and prone to micro-aggregation in solution. The authors perform size exclusion chromatography as a purification step, but none of the chromatograms are shown. Do the domains run as monomers or oligomers? Are there significant aggregate peaks in the chromatograms?*

We have included the chromatograms as a new figure supplement (Figure 6—figure supplement 2). Both inactive proteins – Chi4S and NUDT9H – show a main monomeric peak which can be isolated fairly well, and a smaller peak corresponding to a dimer, just as the highly soluble (and active) NUDT9 protein. Higher-stoichiometry aggregates constitute only a minor fraction in each case. Only the fractions corresponding to the monomeric peaks were collected and pooled for further use in functional assays. Thus, lack of catalytic activity for Chi4S and NUDT9H is not due to protein microaggregation in a quasi-stable solution. We have added this information to the Results section which presents high-resolution colorimetric assays on catalytic activities.

*Are the domains monodispersed when re-run on size exclusion chromatography, or are they only quasi-stable? Do the proteins remain monodispersed after concentration for the binding assays? These questions need to be addressed in order for the conclusion that there is no catalytic conclusion to be certain. While the binding assay in principle could confirm that the domains are well behaved, the weak signal observed in the fluorescent binding assay could be due to non-specific effects of aggregated protein on fluorescence rather than to specific binding. Since BSA is a highly soluble protein, it is not a great control in this case. In summary, the chromatograms should be shown and these points discussed.*

The protein concentrations obtained after pooling monomeric fractions of the SEC run were typically 0.7-1 mg/ml, corresponding to 20-30 μM protein. For the enzymatic assays these preparations were not further concentrated, but rather diluted (to 0.1-1 μM for the TLC-based, and to 5-50 nM for the colorimetric assay). The only assay in which the protein was concentrated following SEC was the fluorescent binding assay (Figure 8): here we had to concentrate Chi4S by ~5-fold, to reach a concentration of 100 μM (3.6 mg/ml). Originally we did not re-run the Chi4S protein on SEC following the binding assay, but we have now performed a control experiment to mimic the effect of this concentration step, and show the results in Figure 8—figure supplement 1. Chi4S was expressed and purified as before (solid blue SEC elution profile), left overnight at 4^o^C, and then concentrated 5-fold using a 10 kDa molecular weight cutoff Vivaspin column (Sartorius). The concentrated protein was then re-run on SEC, and the elution profile (dashed blue profile) again showed a preserved main monomeric peak with a smaller peak corresponding to a dimer. Importantly, larger aggregates did not reappear. This confirms that the signal observed in the fluorescent binding assay is not due to non-specific effects of aggregated protein. We have added this information to the Materials and methods section that describes the binding assay.